# Impact of Tyrosine Kinase Inhibitors on the Expression Pattern of Epigenetic Regulators

**DOI:** 10.3390/cancers17081282

**Published:** 2025-04-10

**Authors:** Klaudia Tóth, Zsuzsanna Gaál

**Affiliations:** Institute of Pediatrics, Faculty of Medicine, University of Debrecen, 4032 Debrecen, Hungary

**Keywords:** tyrosine kinase inhibitors, epigenetics, precision oncology

## Abstract

Despite the emerging possibilities for personalized anticancer treatment, knowledge remains limited regarding the impact of distinct groups of targeted agents on the widespread network of signaling pathways and transcriptional regulatory mechanisms. The primary aim of this study was to evaluate how tyrosine kinase inhibitors (TKIs) influence the expression of epigenetic regulators through the analysis of Gene Expression Omnibus (GEO) datasets. Our results confirm numerous significant changes in the expression levels of epigenetic writers, erasers and readers as a consequence of TKI treatment in both hematological malignancies and solid tumors. These findings could facilitate further translational research aimed at establishing novel treatment combinations with enhanced antitumor efficacy in a personalized approach.

## 1. Introduction

Based on the ten-year age-standardized data of net cancer survival, an increment of more than 25% was registered in England and Wales during 2010–2011 as compared to 1971–1972 [1]. In the United States, 5-year overall survival (OS) of malignant diseases was 70% in 2022 [2]. Regarding childhood cancer, 5-year OS data reached 81% in 2010–2014 in Europe [3]. One of the most important contributors of improved treatment outcome measures is the rapid progress in the methodology of molecular genetics, imaging techniques and minimally invasive interventional techniques [4,5].

The new horizons of targeted treatment that have opened up in recent decades are another promising field for more successful cancer treatment and improved life quality of survivors. Targeted therapeutic agents inhibit important regulatory proteins responsible for the growth and proliferation of tumor cells in a personalized manner [6]. Tyrosine kinase inhibitors (TKIs) are of distinguished significance in precision oncology, and have been successfully used in clinical practice since 2001, when the FDA approved the usage of imatinib mesylate [7]. In 2024, there were over 40 TKI compounds approved by the FDA, including BCR-ABL, JAK2, CDK, Btk, EGFR, MEK, BRAF and ALK inhibitors [8].

The two main classes of tyrosine kinase enzymes are receptor tyrosine kinases (RTK) and cytoplasmic tyrosine kinases, catalyzing the transfer of ATP-derived phosphate groups to the tyrosine residues of target proteins [9]. While RTKs are a group of transmembrane receptors [6], cytoplasmic tyrosine kinases contain an SH2-domain that mediates substrate recruitment and subcellular localization of the enzymes [10]. TKI agents can be classified into five major subtypes. While type I TKIs are ATP-competitive inhibitors, agents of the type II group are characterized by a binding site adjacent to the ATP site, thereby maintaining inactive conformation [7]. Type III TKIs are allosteric inhibitors, whereas the type IV subgroup is characterized by reversible targeting of the substrate-binding site [7]. Irreversible covalent binding of the kinase active site is a hallmark of type V TKI agents [11].

Despite the triumph of TKI agents, there are still clinically significant obstacles to overcome. Problems related to toxicity can be classified into two major categories. Excessive inhibition of a distinct tyrosine kinase enzyme results in on-target side effects, while the simultaneous inhibition of other protein kinases can lead to the development of off-target consequences [12]. Cardiac involvement, such as heart failure, pericardial effusion and QT prolongation, accounts for serious complications [12]. Elevations of hepatic aminotransferase enzymes are also frequently reported [13]. Besides toxicity, resistance to TKIs is also a common issue. ABL-dependent pathways include gene amplification and BCR-ABL mutations, while ABL-independent TKI resistance may result from high affinity of serum proteins or differences in oral bioavailability [14].

One of the keys to further improving treatment results may be to combine TKIs with targeted therapeutic agents with other mechanisms of action. The double BRAF-MEK inhibition, recommended in melanoma, targets an RTK and one of its downstream effectors simultaneously [15]. In pediatric BRAFV600E-mutated low-grade glioma, longer progression-free survival can be reached with combined administration of BRAF-inhibitor dabrafenib and MEK-inhibitor trametinib, as compared to standard first-line chemotherapy [16]. Developing combinations of TKIs with immunotherapy is also a promising option. The combinations of atezolizumab (anti-PD-L1) with sunitinib (VEGFR inhibitor), and pembrolizumab (anti-PD1) with dabrafenib (BRAF inhibitor) and trametinib (MEK inhibitor) are under phase 3 and phase 2 clinical trials in metastatic RCC and BRAF-mutant melanoma, respectively [17].

Currently, tyrosine kinase inhibitors are used in combination with other effective therapeutic agents. It should be noted that tyrosine kinase inhibitors with anticancer nanoparticles are quite actively used; for example, the selenium–sorafenib nanocomplex has an anticancer effect on human hepatocyte carcinoma cells, where nanoselenium is an active compound, besides its transporter function [18]. On the contrary, nanoparticles are made for targeted delivery of sorafenib to the tumor; for example, medium-chain triglycerides and lactobionic acid-human serum albumin conjugate (LA-HSA conjugate) are used to create sorafenib-loaded albumin lipid nanoparticles (ALNs) as a targeted drug delivery system to treat hepatocellular carcinoma (HCC) [19].

Arising from recent advances in survival outcome measures of cancer, long-term survivors are highly prevalent. Therefore, efforts to avoid long-term adverse effects and introduce less toxic treatment regimens are of distinguished clinical significance. Within the population of the US aged 40–54 years, 3.6% and 2.1% were cancer survivors in 2022 among females and males, respectively, which ratio increased up to 14.5% in females and 16% in males between 65–74 years of age [2]. In the era of precision oncology, TKI agents became the most widely used targeted therapeutic agents with various indications including leukemia, lymphoma and solid tumors as well. However, dose-limiting side effects and the development of resistance are still major clinical problems. In recent years, fascinating novel combinations of self-tailored smart molecules have been initiated, such as the supplementation of TKI treatment with immunotherapy [17] and epigenetic agents [20].

As a recently recognized hallmark of cancer, epidrugs targeting disrupted epigenetic regulation are increasingly used in antitumor treatment schedules. In a KRAS-mutated NSCLC cell line, the combination of erlotinib with histone deacetylase inhibitors (HDACi) enhanced the apoptosis of cancer cells [20]. Similarly, TKI combined with venetoclax (BCL2 inhibitor) and azacitidine (hypomethylating agent) was found to be an effective and safe regimen for patients with de novo lymphoid blast phase-CML [21]. However, few data are available on how TKIs influence the expression level of epigenetic modifiers. Since this can be applied in the development of novel therapeutic combinations, the main goal of our study was to examine how TKI agents affect the expression level of enzymes responsible for epigenetic regulation. Clinical implications of the most-affected epigenetic regulators by TKIs are also discussed in detail in the mirror of our findings.

The main goal of our study was to investigate how TKI agents affect the expression level of epigenetic regulators, a huge group of proteins and non-coding RNA molecules that can be classified as epigenetic writers, erasers, readers, ATP-dependent chromatin remodeling complexes, and non-coding RNA molecules including miRNAs [22] (Table 1).

## 2. Materials and Methods

### 2.1. GEO Database

GEO (Gene Expression Omnibus) is a public database that allows the retrieval of gene expression patterns through microarray studies, next-generation sequencing and other genomic data shared by research teams. In our study, we selected GEO datasets and data series from hematological malignant diseases and solid tumors that were treated with TKI agents either in vitro or in vivo (Table 2). In each dataset, epigenetic regulators were selected from those 200 genes that showed the greatest difference in expression levels before and after TKI treatment. Two additional datasets of untreated tumors were analyzed (GSE37418, GSE28703). Finally, analysis was completed with the evaluation of a dataset obtained from epidrug-treated leukemia cell lines (GSE29828) (Appendix A).

### 2.2. Data Visualization and Statistical Analysis

Statistical analysis of gene expression data and design of figures were performed via GraphPad Prism 8.0.1. software. A significance level of 5% was applied. We assumed a continuous or discrete variable for the datasets. Thus, when comparing more than two groups, the Kruskal–Wallis test was used, otherwise the Mann–Whitney Rank Sum test was performed to establish significant differences between two groups.

## 3. Results

### 3.1. Comparison of Expression Levels of Epigenetic Regulators Among Distinctive Subgroups of Malignant Diseases W/O Treatment

#### 3.1.1. Distinctive Subgroups of a Solid Tumor: GSE37418. Novel Mutations Target Distinct Subgroups of Medulloblastoma

In the case of the GSE37418 dataset, *n* = 73 pediatric medulloblastoma samples were analyzed, representing four expression classes, WNT-activated, SHH-activated, Group 3 and Group 4 medulloblastoma (WNT *n* = 8, SHH *n* = 10, G3 *n* = 16, G4 *n* = 39). miR4454, miR100HG and ASXL3 epigenetic regulators were selected from those 200 genes that showed the greatest difference in expression levels in the case of untreated medulloblastoma subtypes. Statistical analysis of gene expression data and design of figures were performed via GraphPad Prism 8.0.1 using the Kruskal-Wallis test. At a significance level of 5%, the expression of miR-4454 was the highest in the WNT-activated subgroup with a favorable prognosis, while ASXL3, non-catalytic adaptor component of the polycomb repressive deubiquitinase complex [39], had the highest expression level in the high-risk G4 group. These observations are in accordance with recent results that suggest the involvement of down-regulated miR-4454 in early metastasis formation [40] and the oncogenic function of the BRD4/ASXL3/BAP1 axis in small-cell lung cancer (SCLC) [40], respectively (Figure 1a).

#### 3.1.2. Distinctive Subgroups of a Hematological Malignant Disease: GSE28703. Discovery of Novel Recurrent Mutations and Rearrangements in Early T-Cell Precursor Acute Lymphoblastic Leukemia by Whole Genome Sequencing

In the case of the GSE28703 dataset, gene expression profiling was performed on *n* = 52 single-diagnosis tumor samples comparing ETP and non-ETP ALL (ETP ALL *n* = 12, non-ETP ALL *n* = 40). SMARCA2, HDAC9 and CITED2 epigenetic regulators were selected from those 200 genes that showed the greatest difference in expression levels in the case of comparing ETP and non-ETP ALL. Early T-cell precursor (ETP)-ALL is a high-risk type of acute leukemia with immature T cells, that share similarities with myeloid progenitor cells and hematopoietic stem cells (HSC) in terms of gene expression profile [41,42]. Statistical analysis of gene expression data and design of figures were performed via GraphPad Prism 8.0.1 using the Mann–Whitney test. At a significance level of 5%, we observed higher expression of SMARCA2 (member of SWI/SNF family of proteins), HDAC9 (histone deacetylase enzyme) and CITED2 (Cbp/p300-interacting transactivator 2) in the ETP group as compared to non-ETP patients. Although none of these have known clinical significance, the higher expression of HDAC7 and HDAC9 enzymes were associated with unfavorable prognosis in pediatric ALL [43], SMARCA2 has important function in the maintenance of HSCs [44], and CD34+ cells from AML patients displayed increased expression levels of CITED2 [45] (Figure 1b).

### 3.2. Comparison of Expression Levels of Epigenetic Regulators Among Distinctive Subgroups TKI-Treated and Untreated Subgroups of Malignant Diseases

#### 3.2.1. Solid Tumors

##### GSE66346: Expression Data from Renal Cancer Xenograft Tumor Treated with Sunitinib or Vehicle

In the case of the GSE66346 dataset, total RNA was extracted from KURC1,2,3 xenograft tumors treated with sunitinib or vehicle compared with untreated ones (treated *n* = 5, untreated *n* = 4). SMARCAL2, miR200C, miR323A, miR382, miR516B1 and miR664B epigenetic regulators were selected from those 200 genes that showed the greatest difference in expression levels in the case of comparing sunitinib-treated and untreated renal cancer xenograft. Statistical analysis of gene expression data and design of figures were performed via GraphPad Prism 8.0.1 using the Mann–Whitney test. At a significance level of 5%, SMARCAL1 (member of SWI/SNF family of proteins) had significantly higher levels, while miR-200c, miR-323a, miR-382, miR-516b and miR-664b had significantly lower levels in the sunitinib-treated group than in the non-treated tumors. Among these miRNAs, miR-200c was associated with lower risk of relapse [46], while sponging of miR-382 by the circular RNA circ-SAR1A promoted the growth and invasion of RCC cells [47]. A homozygous missense variant of the SMARCAL1 gene was described in two siblings with Schimke immuno-osseous dysplasia, one of whom developed a classical type of congenital mesoblastic nephroma combined with disturbed immunoglobulin production and profound lymphopenia as well [48] (Figure 2a).

##### GSE197555: Differential mRNA Expression Analysis of H460 Cells and A549 Cells After Trametinib Treatment

In the case of the GSE197555 dataset, transcription profiling of lung cancer cells was done after trametinib treatment comparing with untreated cases (treated *n* = 6, untreated *n* = 6). CHD4, DPY30 and miR614 epigenetic regulators were selected from those 200 genes that showed the greatest difference in expression levels in the case of comparing trametinib-treated and untreated NSCLC H460 and A459 cells. Statistical analysis of gene expression data and design of figures were performed via GraphPad Prism 8.0.1 using the Mann–Whitney test. At a significance level of 5%, trametinib treatment resulted in significantly different expression levels of CHD4 (core nucleosome-remodeling component of NuRD complex), DPY30 (regulatory subunit of H3K4 histone methyltransferase complex) and miR-614 as compared to the non-treated cells. MiR-614 was found to inhibit cell proliferation [49], CHD4 regulates both proliferative and migratory abilities of NSCLC cells via the RhoA/ROCK pathway [50], while the role of DPY30 has not yet been clarified in lung cancer (Figure 2b).

##### GSE59357: Gene Expression Profiles of Dasatinib-Resistant and Dasatinib-Sensitive Pancreatic Cancer Cell Lines

In the case of the GSE59357 dataset, RNA from three dasatinib-resistant (MiaPaCa2, Panc1, SU8686) and three dasatinib-sensitive (Panc0504, Panc0403, Panc1005) pancreatic cancer cell lines were extracted (sensitive *n* = 9, resistant *n* = 9). CITED2, SMARCD3 and miR132 epigenetic regulators were selected from those 200 genes that showed the greatest difference in expression levels in the case of comparing dasatinib-sensitive and dasatinib-resistant pancreatic cancer cells. Statistical analysis of gene expression data and design of figures were performed via GraphPad Prism 8.0.1 using the Mann–Whitney test. At a significance level of 5%, the comparison revealed significantly higher CITED2 and SMARCD3 (member of SWI/SNF family of proteins) expression, and lower miR-132 level in dasatinib-resistant cells. SWI-/SNF-deficient pancreatic undifferentiated carcinoma was associated with worse overall survival and highly aggressive behavior [51]. Proliferation and survival of pancreatic beta cells is regulated by miR-132 through PTEN/Akt/Foxo3 signaling and caspase 9 [52] (Figure 2c).

##### GSE42872: Expression Data from BRAFV600E A375 Melanoma Cells Treated with Vehicle or Vemurafenib

In the case of the GSE42872 dataset, BRAFV600E A375 human melanoma cells were treated with vehicle (0.1% DMSO) or 10 μM vemurafenib for 24 h after which total RNA was extracted (treated *n* = 3, untreated *n* = 3). HIST1H1A, HIST1H1B, HIST1H2AB, HIST1H2BB, HIST1H2BF, HIST1H3A, KAT2B, UHRF1, miR17HG and miR221 epigenetic regulators were selected from those 200 genes that showed the greatest difference in expression levels in the case of comparing vemurafenib-treated and untreated melanoma cells. Statistical analysis of gene expression data and design of figures were performed via GraphPad Prism 8.0.1 using the Mann–Whitney test. Although at a significance level of 5%, no statistically significant differences were found in the expression of epigenetic regulators following vemurafenib treatment; however, miR-221 and UHRF1 (binds hemimethylated DNA and recruits DNMT1) were among the most differentially expressed genes, which were described as biomarkers of nevi/melanoma [53] and regulators of melanocytic differentiation [54], respectively (Appendix A).

##### GSE98314: Melanoma Cell Lines Treated with Dabrafenib ± Trametinib

In the case of the GSE98314 dataset, eleven melanoma cell lines were grown in DMEM/10%FBS and treated with either 100 nM dabrafenib or DMSO for 24 h (dabrafenib-treated *n* = 7, untreated *n* = 11). HIST1H4C and UHRF1 epigenetic regulators were selected from those 200 genes that showed the greatest difference in expression levels in the case of comparing dabrafenib-treated and -untreated melanoma cells. Statistical analysis of gene expression data and design of figures were performed via GraphPad Prism 8.0.1 using the Mann–Whitney test. At a significance level of 5%, UHRF1 and HIST1H4C showed significantly higher expression level in non-treated cells as compared to dabrafenib treatment. Although the clinical significance of these two genes is unknown, selective inhibition of HDAC6 promoted apoptosis and enhanced chemosensitivity of melanoma cells [55] (Figure 2d).

#### 3.2.2. Leukemias and Lymphomas

##### GSE23743: Effect of Imatinib on Philadelphia Chromosome Positive Acute Lymphoblastic Leukemia

In the case of the GSE23743 dataset, four Ph+ ALL cell lines (BV-173, NALM-1, SUP-B15, and TOM1) were either treated with 10µM STI571 (Imatinib) for 16 h or cultured in absence of STI571 (treated *n* = 4, untreated *n* = 4). GCNT1P1, KDM5B/JARID1B (lysine demethylase 5B), SETD2 (H3K36 lysine methyltransferase), MeCP2 (binds methylated CpGs), PHF21A (component of a corepressor complex), miR-3652 and miR-6733 were the epigenetic regulators listed among the 200 genes with the most different expression levels after imatinib treatment in Ph+ ALL cell lines. Statistical analysis of gene expression data and design of figures were performed via GraphPad Prism 8.0.1 using the Mann–Whitney test. Although at a significance level of 5%, none of the differences were statistically significant, their clinical impact may yet be considerable. KDM5B/JARID1B is repressed by IKZF1 transcription factor, alterations in which are strongly associated with Ph+ ALL. Therefore, KDM5B is a recently identified therapeutic target in B-ALL, the impaired repression of which could be restored by the targeted inhibition of casein kinase 2 (CK2) [56]. Inactivating SETD2 mutation was described as a mechanism of chemoresistance in leukemia. Sensitivity to cytarabine in such cases can be restored by inhibition of the H3K36 demethylase KDM4A enzyme [57] (Appendix A).

##### GSE24493: Effect of Imatinib on Chronic Myelogenous Leukemia

In the case of the GSE24493 dataset, three CML cell lines (KU-812, KCL-22, JURL-MK1) were either treated with 10 µM STI571 (Imatinib) for 16 h or cultured in absence of STI571 (treated *n* = 3, untreated *n* = 3). BAZ1A, BAZ2B, CHAF1A, DOT1L, JMJD1C and miR612 epigenetic regulators were selected from those 200 genes that showed the greatest difference in expression levels in the case of comparing imatinib-treated and untreated CML cells. Statistical analysis of gene expression data and design of figures were performed via GraphPad Prism 8.0.1 using the Mann–Whitney test. At a significance level of 5%, similarly to Ph+ ALL cell lines, no statistically significant differences in the expression levels of epigenetic regulators have been confirmed during the analysis. However, DOT1L (H3K79 methyltransferase) and JMJD1C (H3K9 demethylase) enzymes were included among those 200 genes that showed the greatest difference in expression following TKI treatment. Though DOT1L has not yet been implicated in CML, its significant role in the development of MLL -rearranged (MLL-r) leukemia is supported by accumulated evidence [58]. Moreover, significant synergy has been confirmed between the DOT1L inhibitor pinometostat and the menin inhibitor revumenib in the treatment of MLL-r ALL [59]. JMJD1C is required for the self-renewal of leukemia stem cells (LSCs) of AML but not normal HSCs [60], and it has been implicated in the function of LSCs in MLL-AF9-driven leukemia, too [61]. Since it was found to be overexpressed in patients with myeloproliferative neoplasms, JMJD1C is considered as a novel therapeutic target in these malignancies [61] (Appendix A).

##### GSE218183: Bulk RNA-Seq Analysis of Primary CML CD34+ Cells (*n* = 3) Treated with Idasanutlin Alone or in Combination with Nilotinib In Vitro

In the case of the GSE218183 dataset, primary CD34+ CML cells isolated from peripheral blood of three unrelated chronic phase CML patients were treated with nilotinib (3 µM), idasanutlin (166 nM), or a combination for 24 hrs or 72 hrs; thereafter, RNA-seq analysis was performed (nilotinib treated *n* = 3, idasanutlin treated *n* = 2, idasanutlin + nilotinib treated *n* = 3, untreated *n* = 3). miR3652, H4C1 and HDAC1 epigenetic regulators were selected from those 200 genes that showed the greatest difference in expression levels in the case of comparing nilotinib- and idasanutlin-treated and untreated CML cells. Statistical analysis of gene expression data and design of figures were performed via GraphPad Prism 8.0.1 using the Kruskal–Wallis test. At a significance level of 5%, we observed significantly higher expression levels of miR-3652 following idasanutlin treatment as compared to control cells. On the other hand, nilotinib treatment resulted in a significant decrement of miR-3652 expression. Therefore, it is suggested that different TKI agents may induce opposite changes in the expression levels of miRNAs (Figure 3a). Similarly, we have found that nilotinib increased, while idasanutlin decreased, the level of the HDAC1 enzyme; however, these differences were not statistically significant. Interestingly, the regulation of HDAC1 is of particular importance for the survival of K562 CML cells, and panobinostat (non-selective HDAC inhibitor) induced apoptosis in both imatinib-sensitive and -resistant K562 cells [62]. Therefore, modulating the activity of HDAC enzymes may be of therapeutic benefit in TKI-resistant CML (Appendix A).

##### GSE197811: Effect of the Janus Kinase Inhibitor Ruxolitinib on Gene Expression of Chronic Lymphocytic Leukemia Cells In Vivo

In the case of the GSE197811 dataset, gene expression in circulating CLL cells before and 4–8 weeks after treatment with ruxolitinib were compared by RNA sequencing (treated *n* = 6, untreated *n* = 6). KDM1A, PHF13 and miR6859 epigenetic regulators were selected from those 200 genes that showed the greatest difference in expression levels in the case of comparing ruxolitinib-treated and untreated CLL cells. Statistical analysis of gene expression data and design of figures were performed via GraphPad Prism 8.0.1 using the Mann–Whitney test. Although at a significance level of 5%, no statistically significant differences were confirmed in the expression levels of epigenetic regulators when comparing treated and untreated groups, but we mention that the level of KDM1A/LSD1 (lysine-specific histone demethylase 1A) decreased upon ruxolitinib treatment, which enzyme has been identified as an oncogenic protein in this disease [63]. PHF13, a reader protein that binds H3K4me3-containing chromatin, also showed lower expression levels in the ruxolitinib-treated group. Though its function is not known in leukemia and lymphoma, PHF13 regulates genes critical for pancreatic cancer cell migration and invasion [64] (Appendix A).

##### GSE171763: Inhibitors of Bcl-2 and Bruton’s Tyrosine Kinase Synergize to Abrogate Diffuse Large B-Cell Lymphoma (DLBCL) Growth

In the case of the GSE171763 dataset, DLBCL cell lines RIVA, U-2932 and OCI-LY3 were treated in vitro with ibrutinib (Ibru) or dimethyl sulfoxide (DMSO); additionally, RIVA and U-2932 were orthotopically transplanted into MISTRG mice and mice were treated for two weeks with Ibru or vehicle, after which total RNA was isolated by next-generation sequencing (treated *n* = 15, untreated *n* = 9). DLBCL is the most common type of non-Hodgkin lymphoma [65]. PHF6, H2BC20P and JMJD6 epigenetic regulators were selected from those 200 genes that showed the greatest difference in expression levels in the case of comparing ibrutinib-treated and untreated DLBCL cells. Statistical analysis of gene expression data and design of figures were performed via GraphPad Prism 8.0.1 using the Mann–Whitney test. At a significance level of 5%, ibrutinib (inhibitor of Bruton’s tyrosine kinase) treatment resulted in a significant decrement of JMJD6 (arginine demethylase and lysine hydroxylase) and PHF6 expression (regulator of chromatin remodeling) in cell lines and a xenotransplantation model of DLBCL. Although no translational clinical applications of either JMJD6 or PHF6 in DLBCL have been established to date, JMJD6 is involved in the maintenance and multilineage differentiation potential of HSCs [66], while the oncogenic effect of PHF6 is suggested in precursor B-cell lymphoblastic leukemia [67] (Figure 3b).

##### GSE173306: Transcription Profiling of ALK-Rearranged Cell Lines Resistant and Sensitive to Crizotinib

In the case of the GSE173306 dataset, seven ALK rearranged cell lines, which became resistant to crizotinib, were compared at transcription level with the corresponding untreated cell lines (treated *n* = 7, untreated *n* = 7). SRSF6, AGO4, BRWD3, KDM6B, PHF21A and BRWD1 epigenetic regulators were selected from those 200 genes that showed the greatest difference in the case of comparing crizotinib-treated and -untreated ALCL cells. Statistical analysis of gene expression data and design of figures were performed via GraphPad Prism 8.0.1 using the Mann–Whitney test. At a significance level of 5%, we observed significantly higher levels of SRSF6 in the crizotinib-treated group as compared to the crizotinib-untreated type. However, the analysis revealed significantly decreased expression of AGO4 (component of miRNA effector complex), BRWD3 (reader protein with bromodomains and WD-repeats), KDM6B (H3K27me3 demethylase) and PHF21A following crizotinib treatment as compared to untreated cells. The role of KDM6B in ALCL is currently unknown. However, inhibition of the KDM6B enzyme resulted in enhanced chemosensitivity of DLBCL cells [68]. In addition, KDM6B has been identified as a novel therapeutic target in NOTCH1-driven T-ALL [69] (Figure 3c). However, at a significance level of 5%, no statistically significant difference was observed in the values of the BRWD1 epigenetic regulator when comparing the treated and untreated ALCL subgroups. Though the function of BRWD3 in hematological malignancies has not been revealed yet, BRWD1 is considered as a master orchestrator of late B-cell development [70] (Appendix A).

##### GSE29828: Expression Profiling of Mixed Lineage Leukemia Cells Treated with a Potent Small-Molecule DOT1L Inhibitor

In the case of the GSE29828 dataset, MV4-11 cells were treated with 3 uM EPZ004777 (DOT1L inhibitor) or vehicle control (0.1% DMSO) for 2, 4 and 6 days; for each unique condition, three biological replicates were generated for expression profiling (treated *n* = 12, untreated *n* = 6). The 200 most differentially expressed genes involved numerous epigenetic regulators including AGO1, AGO4, JMJD6, miR-15a, miR-6758, miR-6787, miR-6836 and miR-6890. The translational clinical applications of oncomiRs, tumor-suppressor anti-oncomiRs and hematopoiesis-regulating miRNAs are continuously expanding in AML, including prognostic biomarkers and miRNA-directed treatment options [71]. MiR-15a is a well-known anti-oncomiR in CLL targeting BCL2 [72]. However, due to its inhibitory effect on daunorubicin-induced autophagy, miR-15a has recently been identified as a promising therapeutic target in chemoresistant AML [73]. Statistical analysis of gene expression data and design of figures were performed via GraphPad Prism 8.0.1 using the Mann–Whitney test. At a significance level of 5%, we observed significantly higher levels of AGO1, AGO4, SRSF8 and miR6890 in the case of the DOT1L inhibitor-treated subgroup as compared to the untreated type. On the contrary, the values of JMJD6, miR15A, miR6758, miR6787 and miR6836 were significantly higher in the DOT1L-inhibitor-untreated group comparing with the treated subtype. According to recent data, evaluation of JMJD6 expression can be applied in the prediction of resistance of AML cells to BET inhibitors [74] (Figure 4).

### 3.3. Summary of Epigenetic Regulators Modulated by TKI Treatment

GEO datasets from TKI-treated hematological malignancies and solid tumors have been analyzed, searching for epigenetic regulators among those 200 genes that showed the greatest difference in expression levels before and after TKI treatment (Table 3). Here, we summarize our findings in the mirror of recent publications on the implication of distinctive epigenetic regulators in malignant diseases. In order to compare the expression profile of epigenetic regulators in distinctive subsets of malignant diseases, one dataset from solid tumors and one from leukemias have initially been analyzed, which were not previously treated with TKI agents.

## 4. Discussion

The main goal of our study was to investigate how TKI agents affect the expression level of epigenetic regulators. GEO datasets from TKI-treated hematological malignancies and solid tumors have been analyzed, searching for epigenetic regulators among those 200 genes the expression of which showed the greatest difference in terms of fold change upon TKI treatment. Analyzed GEO datasets included hematological malignancies (e.g. Ph+ ALL, CML, CLL, lymphomas) and a wide variety of solid tumors as well. To date, more than 40 TKI agents have been approved by FDA and are successfully applied in a wide variety of hematological malignancies and solid tumors as well. Although they are generally well-tolerated and effective, major clinical issues, such as the development of resistance, pericardial/pleural effusions and hepatotoxicity still need to be considered. Based on data in the literature, the majority of these regulators either play a role in the pathogenesis of the disease or can be used to predict chemoresistance or prognosis, indicating a continuously growing number of novel therapeutic targets.

Despite several limitations, such as the limited number of datasets, the analysis resulted in inspiring new observations. Statistically significant differences have been found in several case of epigenetic regulators comparing the TKI-treated and untreated groups. Our most important finding is that TKI agents considerably influence the expression of epigenetic writers, erasers (such as JMJD6), miRNAs (e.g.: miR200C, miR614, miR132) and members of chromatin-remodeling complexes (e.g.: SMARCAL1), too. These differences support the previous findings that epigenetic regulators play an important role in tumorigenesis and treatment, as well. According to the literature, there are completed or ongoing clinical trials to examine drugs affecting these epigenetic regulators.

According to recent data, novel therapeutic combinations of TKIs may further improve treatment results, among which epigenetic agents are of special interest. One of the keys to further improving treatment results may be to combine TKIs with targeted therapeutic agents with other mechanisms of action. The combination of the VEGF inhibitor bevacizumab with the EGFR inhibitor erlotinib in recurrent/metastatic squamous cell carcinoma of the head and neck is under phase 1 and 2 clinical trials [75]. Developing combinations of TKIs with immunotherapy is also a promising option. The combinations of atezolizumab (anti-PD-L1) with bosutinib (Abl-Src inhibitor) in CML is under phase 1-2 clinical trial (NCT04793399), and the combination of G-CSF (sargramostim) with nilotinib (BCR-Abl inhibitor) in ALL is under phase 3 clinical trial (NCT02611492) [76].

## 5. Conclusions

In conclusion, expression changes of epigenetic regulators described in our study in TKI-treated malignancies may facilitate in vitro experiments and clinical trials in which novel treatment combinations of TKIs and epidrugs can be established.

## Figures and Tables

**Figure 1 cancers-17-01282-f001:**
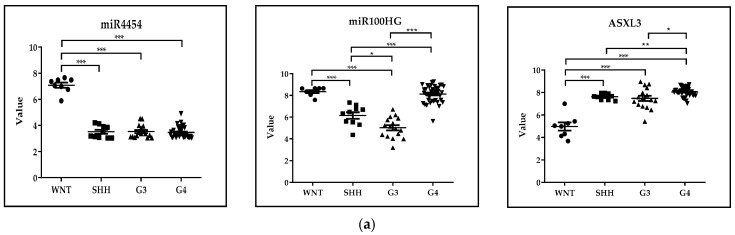
(**a**) **GSE37418**. *Novel mutations target distinct subgroups of medulloblastoma*. miR4454, miR100HG and ASXL3 epigenetic regulators were selected from those 200 genes that showed the greatest difference in expression levels in the case of untreated medulloblastoma subtypes. Statistical analysis was performed using the Kruskal–Wallis test. At a significance level of 5%, we observed significant differences in the values of these epigenetic regulators in several cases when comparing the individual medulloblastoma subgroups. *** *p* < 0.001, ** *p* < 0.01, * *p* < 0.05; WNT *n* = 8, SHH *n* = 10, G3 *n* = 16, G4 *n* = 39. (**b**) **GSE28703***. Discovery of novel recurrent mutations and rearrangements in early T-cell precursor acute lymphoblastic leukemia by whole genome sequencing.* SMARCA2, HDAC9 and CITED2 epigenetic regulators were selected from those 200 genes that showed the greatest difference in expression levels in the case of comparing ETP and non-ETP ALL. Statistical analysis was performed using the Mann–Whitney test. At a significance level of 5%, we observed significant differences in the values of these epigenetic regulators when comparing the ALL subgroups. *** *p* < 0.001, ETP-ALL *n* = 12, non-ETP ALL *n* = 40.

**Figure 2 cancers-17-01282-f002:**
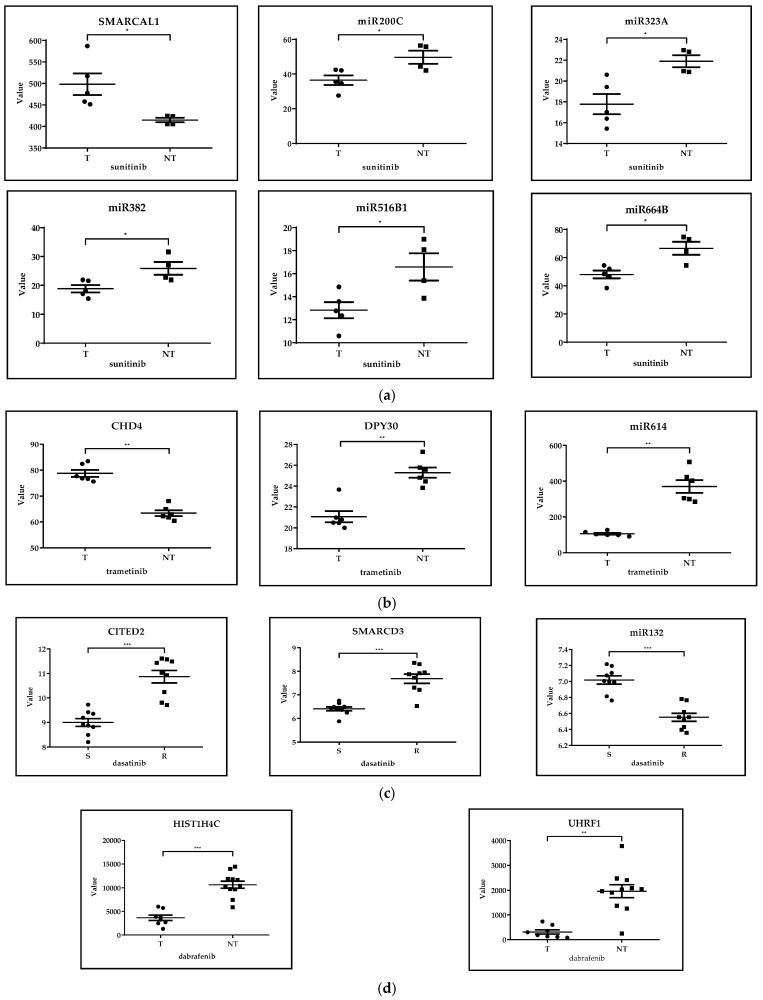
(**a**) **GSE66346.**
*Expression data from renal cancer xenograft tumor treated with sunitinib or vehicle.* SMARCAL2, miR200C, miR323A, miR382, miR516B1 and miR664B epigenetic regulators were selected from those 200 genes that showed the greatest difference in expression levels in the case of comparing sunitinib-treated and untreated renal cancer xenograft. Statistical analysis was performed using the Mann–Whitney test. At a significance level of 5%, we observed significant differences in the values of these epigenetic regulators when comparing the treated and untreated renal cancer subgroups. * *p* < 0.05, T = treated *n* = 5, NT = not treated *n* = 4. (**b**) **GSE197555***. Differential mRNA expression analysis of H460 cells and A549 cells after trametinib treatment.* CHD4, DPY30 and miR614 epigenetic regulators were selected from those 200 genes that showed the greatest difference in expression levels in the case of comparing trametinib-treated and untreated NSCLC H460 and A459 cells. Statistical analysis was performed using the Mann–Whitney test. At a significance level of 5%, we observed significant differences in the values of these epigenetic regulators when comparing the treated and untreated NSCLC subgroups. ** *p* < 0.01, T = treated *n* = 6, NT = not treated *n* = 6. (**c**) **GSE59357**. *Gene expression profiles of dasatinib-resistant and dasatinib-sensitive pancreatic cancer cell lines.* CITED2, SMARCD3 and miR132 epigenetic regulators were selected from those 200 genes that showed the greatest difference in expression levels in the case of comparing dasatinib-sensitive and dasatinib-resistant pancreatic cancer cells. Statistical analysis was performed using the Mann–Whitney test. At a significance level of 5%, we observed significant differences in the values of these epigenetic regulators when comparing the sensitive and resistant pancreatic cancer subgroups, *p* < 0.001, S = sensitive *n* = 9, R = resistant *n* = 9. (**d**) **GSE98314**. *Melanoma cell lines treated with dabrafenib ± trametinib*. HIST1H4C and UHRF1 epigenetic regulators were selected from those 200 genes that showed the greatest difference in expression levels in the case of comparing dabrafenib-treated and untreated melanoma cells. Statistical analysis was performed using the Kruskal–Wallis test. At a significance level of 5%, we observed significant differences in the values of these epigenetic regulators when comparing the treated and untreated melanoma subgroups, *** *p* < 0.001, ** *p* < 0.01, T = treated with dabrafenib *n* = 7, NT = not treated *n* = 11.

**Figure 3 cancers-17-01282-f003:**
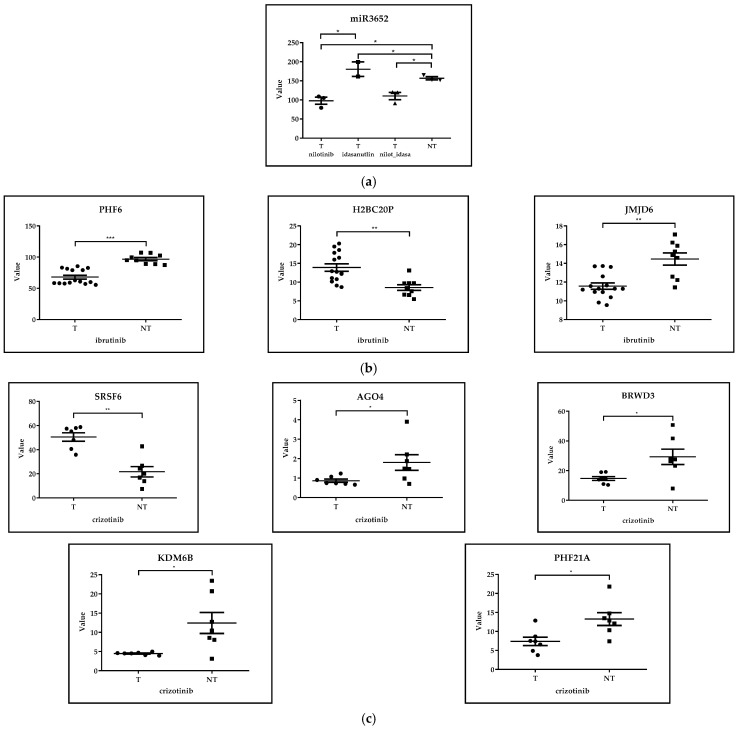
(**a**) **GSE218183.** Bulk RNA-seq analysis of primary CML CD34+ cells (*n* = 3) treated with idasanutlin alone or in combination with nilotinib in vitro. miR3652, H4C1 and HDAC1 epigenetic regulators were selected from those 200 genes that showed the greatest difference in expression levels in the case of comparing nilotinib- and idasanutlin-treated and untreated CML cells. Statistical analysis was performed using the Kruskal-Wallis test. At a significance level of 5%, we observed significant differences in the values of the miR3652 epigenetic regulator when comparing the treated and untreated CML cells subgroups. * *p* < 0.05, T = treated nilotinib *n* = 3, idasanutlin *n* = 2, nilotinib plus idasanutlin *n* = 3, NT = not treated *n* = 3, ns = not significant. (**b**) **GSE171763.** Inhibitors of Bcl-2 and Bruton’s tyrosine kinase synergize to abrogate diffuse large B-cell lymphoma (DLBCL) growth. PHF6, H2BC20P and JMJD6 epigenetic regulators were selected from those 200 genes that showed the greatest difference in expression levels in the case of comparing ibrutinib-treated and untreated DLBCL cells. Statistical analysis was performed using the Mann–Whitney test. At a significance level of 5%, we observed significant differences in the values of these epigenetic regulators when comparing the treated and untreated DLBCL cells subgroups. *** *p* < 0.001, ** *p* < 0.01. T = treated *n* = 15, NT = not treated *n* = 9. (**c**) **GSE173306.** Transcription profiling of ALK-rearranged cell lines resistant and sensitive to crizotinib. SRSF6, AGO4, BRWD3, KDM6B, PHF21A and BRWD1 epigenetic regulators were selected from those 200 genes that showed the greatest difference in expression levels in the case of comparing crizotinib-treated and untreated ALCL cells. Statistical analysis was performed using the Mann–Whitney test. At a significance level of 5%, we observed significant differences in the values of SRSF6, AGO4, BRWD3, KDM6B and PHF21A epigenetic regulators when comparing the treated and untreated ALCL cells subgroups. ** *p* < 0.01, * *p* < 0.05, T = treated *n* = 7, NT = not treated *n* = 7.

**Figure 4 cancers-17-01282-f004:**
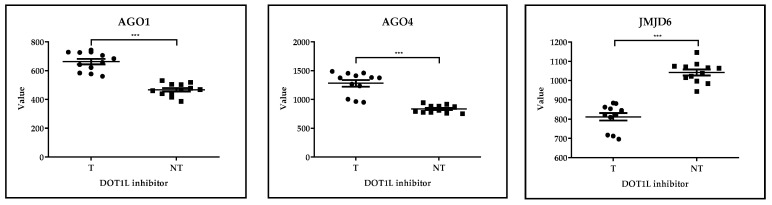
**GSE29828.** *Expression Profiling of Mixed Lineage Leukemia Cells Treated with a Potent Small-Molecule DOT1L Inhibitor.* AGO1, AGO4, JMJD6, SRSF8, miR15A, miR6758, miR6787, miR6836 and miR6890 epigenetic regulators were selected from those 200 genes that showed the greatest difference in expression levels in the case of comparing DOT1L inhibitor-treated and untreated AML cells. Statistical analysis was performed using the Mann–Whitney test. At a significance level of 5%, we observed significant differences in the values of these epigenetic regulators when comparing the treated and untreated AML subgroups. *** *p* < 0.001, T = treated *n* = 12, NT = not treated *n* = 6.

**Table 1 cancers-17-01282-t001:** Main groups of epigenetic regulators.

**Writer**	**Eraser**	**Reader**	**ncRNA**	**Chromatin Remodeling**
***DNA methyltransferase*** DNMT ***DNA hydroxymethylase*** TET ***Histone acetyltransferase*** CBP ELP3 ESA1 Gcn HAT HBO1 HPA2 MOF MORF MOZ p300 PCAF Sas Tip60 YBF ***Histone methyltransferase*** ASH1L Bmi1 (ub) DOT1L EHMT EZH2 G9a GLP KMT MLL NSD PRDM PRMT SETD SUV39H SUV420H	***Histone demethylase*** JARID JHDM JMJD KDM LSD PHF UTF ***Histone deacetylase*** HDAC SIRT	AIRE ASH ATRX BAZ BHC80 BRD BRPF1 CHD Cps35 EAF3 EED ING2 MBD MDC1 MeCP PB1 SP1 SUZ TAF TDRD3 TRIM	LncRNA microRNA miRNA piRNA ***Components of RISC complexes*** AGO DICER Drosha	BAF BRG BRM CHRAC hACF1 Ini1 INO80 ISWI Mi-2 NCoR NURD PRC SMARC Snf Swi Swp SWR1

**Table 2 cancers-17-01282-t002:** GEO datasets.

Dataset Number	Title	Citations
**GSE37418**	Novel mutations target distinct subgroups of medulloblastoma.	[23,24]
**GSE28703**	Discovery of novel recurrent mutations and rearrangements in early T-cell precursor acute lymphoblastic leukemia by whole genome sequencing.	[25,26]
**GSE66346**	Expression data from renal cancer xenograft tumor treated with sunitinib or vehicle.	[27]
**GSE197555**	Differential mRNA expression analysis of H460 cells and A549 cells after trametinib treatment.	[28]
**GSE59357**	Gene expression profiles of dasatinib-resistant and dasatinib-sensitive pancreatic cancer cell lines.	[29]
**GSE42872**	Expression data from BRAFV600E A375 melanoma cells treated with vehicle or vemurafenib.	[30]
**GSE98314**	Melanoma cell lines treated with dabrafenib ± trametinib.	[31]
**GSE23743**	Effect of imatinib on philadelphia chromosome positive acute lymphoblastic leukemia.	[32]
**GSE24493**	Effect of Imatinib on chronic myelogenous leukemia.	[32,33]
**GSE218183**	Bulk RNA-seq analysis of primary CML CD34+ cells (*n* = 3) treated with idasanutlin alone or in combination with nilotinib in vitro.	[34]
**GSE197811**	Effect of the janus kinase inhibitor ruxolitinib on gene expression of chronic lymphocytic leukemia cells in vivo.	[35]
**GSE171763**	Inhibitors of Bcl-2 and Bruton’s tyrosine kinase synergize to abrogate diffuse large B-cell lymphoma (DLBCL) growth.	[36]
**GSE173306**	Transcription profiling of ALK-rearranged cell lines resistent and sensitive to crizotinib.	[37]
**GSE29828**	Expression Profiling of Mixed Lineage Leukemia Cells Treated with a Potent Small-Molecule DOT1L Inhibitor.	[38]

**Table 3 cancers-17-01282-t003:** Epigenetic regulators for significant differences found in GSE datasets comparing the TKI-treated and untreated groups.

Dataset Number	Diagnosis	TKI	In Vivo/In Vitro	EpigeneticRegulator	Expression Level After TKI Treatment	Statistical Correlation	Clinical Trial Number of the Drug Against the Epigenetic Regulator
GSE66346	renal cancer	sunitinib	in vivo	**SMARCAL1**		** *p* ** **< 0.05**	-
**miR200C**		NCT02579187
**miR323A**		-
**miR382**		-
**miR516B1**		-
**miR664B**		-
GSE197555	NSCLC	trametinib	in vitro	**CHD4**		** *p* ** **< 0.01**	-
**DPY30**		-
**miR614**		-
GSE59357	pancreatic cancer	dasatinib	in vitro	**CITED2**		** *p* ** **< 0.001**	-
**SMARCD3**		-
**miR132**		-
GSE42872	melanoma	vemurafenib	in vitro	HIST1H1A		*p* = 0.100	-
HIST1H1B		-
HIST1H2AB		-
HIST1H2BB		-
HIST1H2BF		-
HIST1H3A		-
KAT2B		-
UHRF1		-
miR17HG		-
miR221		***LNA***-***i**-**miR**-**221***/NCT04811898/
GSE98314	melanoma	dabrafenib, trametinib	in vitro	**HIST1H4C**		** *p* ** **< 0.001**	-
**UHRF1**		** *p* ** **< 0.01**	-
GSE23743	Philadelphia positive ALL	imatinib	in vitro	GCNT1P1		*p* = 0.125	-
KDM5B		-
MECP2		***trofinetide***/NCT04181723/
PHF21A		-
SETD2		***AZD1775***/NCT03284385/***axitinib***/NCT05941637/
miR3652		-
miR6733		-
GSE24493	CML	imatinib	in vitro	BAZ1A		*p* = 0.100	-
BAZ2B		-
CHAF1A		-
DOT1L		***EPZ5676***/NCT02141828/***pinometostat***/NCT03701295/
JMJD1C		-
miR612		-
GSE218183	CML	idasanutlin, nilotinib	in vivo	**miR3652**	 (n)  (i)  (n&i)	** *p* ** **< 0.05**	**-**
H4C1	 (n)  (i)  (n&i)	*p* = 0.1087	**-**
HDAC1	 (n)  (i)  (n&i)	*p* = 0.0764	***tucidinostat***/NCT05320640/***chidamide***/NCT04514081 NCT04233294 NCT06393361 NCT06563778/
GSE197811	CLL	ruxolitinib	in vivo	KDM1A		*p* = 0.4225	***IMG***-***7289***/NCT04081220/
PHF13		*p* = 0.4848	** - **
miR6859		*p* = 0.6991	** - **
GSE171763	DLBCL	ibrutinib	in vitro *et* in vivo	**PHF6**		** *p* ** **< 0.0001**	-
**H2BC20P**		** *p* ** **< 0.001**	-
**JMJD6**		** *p* ** **< 0.001**	-
GSE173306	ALCL	crizotinib	in vitro	**SRSF6**		** *p* ** **< 0.01**	-
**AGO4**		** *p* ** **< 0.05**	-
**BRWD3**		** *p* ** **< 0.05**	-
**KDM6B**		** *p* ** **< 0.05**	-
**PHF21A**		** *p* ** **< 0.05**	-
BRWD1		*p* = 0.0842	-
GSE29828	AML	DOT1L inhibitor	in vitro	**AGO1**		** *p* ** **< 0.001**	-
**AGO4**		-
**JMJD6**		-
**SRSF8**		-
**miR15A**		-
**miR6758**		-
**miR6787**		-
**miR6836**		-
**miR6890**		-

## Data Availability

The original contributions presented in this study are included in the article and Appendix A. Further inquiries can be directed to the corresponding author.

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
