# Peer review of "Impact of Tyrosine Kinase Inhibitors on the Expression Pattern of Epigenetic Regulators"

_cancers, 2025, doi:10.3390/cancers17081282_

Round 1
Reviewer 1 Report
Comments and Suggestions for Authors
Review of the manuscript "Impact of tyrosine kinase inhibitors on the expression pattern of epigenetic regulators" by Klaudia Tóth and Zsuzsanna Gaá. The manuscript begins with an introduction written in clear and understandable language. The authors review the progress achieved in the treatment of oncological diseases over the past 20 years. Indeed, tyrosine kinase inhibitors have revolutionized. The median survival time has increased significantly, I know this not only from the literature, but also from the example of my family. Currently, tyrosine kinase inhibitors are used in combination with other effective therapeutic agents. It should be noted that tyrosine kinase inhibitors with anticancer nanoparticles are quite actively used, for example 10.3390/ijms23126641. Or, on the contrary, nanoparticles are made for targeted delivery of sorafenib to the tumor, for example 10.1016/j.jddst.2022.103142. All this needs to be taken into account in the manuscript. The materials and methods section is written quite clearly, they took other people's data and analyzed it. The results section of the manuscript begins ambiguously. In section 3.1.1. there are references to literature in the title. How to understand this? Then the authors step by step consider each individual case from the database and build a visual graph based on the known data, and so on 14 times. After this, the manuscript contains a huge section of the discussion manuscript. The first two paragraphs are devoted to general information, then there are 3 tables: one with the names of the genes, the second and third with the levels of significance. Then there are 3 pages of text that have little to do with the information obtained in the results section… The main discovery of the manuscript is that tyrosine kinase inhibitors significantly affect the expression of epigenetic writers, erasers, miRNAs and members of chromatin remodeling complexes. Almost all tyrosine kinase inhibitors provide a decrease in tumor cell proliferation, which suggests that the authors have made the right conclusions, even if they were too expected… Minor points 1. The manuscript states that this is a review, but it is a regular article. 2. The manuscript is submitted to the journal Cancer, while the file I am reviewing says the journal Biomedicine. 3. Figures should be entered into the text after the first mention, and not after 3 pages. Constantly flipping back and forth through the manuscript is inconvenient! 4. in vitro / in vivo and other Latin are written in italics
Author Response
We would like to sincerely thank the Reviewer for the insightful and thought-provoking comments. We fully agree with these observations, and to the best of our ability, we have incorporated the suggestions into the revised manuscript. We believe that these changes have made the manuscript more substantial and improved its overall quality.
Comments 1: Review of the manuscript "Impact of tyrosine kinase inhibitors on the expression pattern of epigenetic regulators" by Klaudia Tóth and Zsuzsanna Gaá. The manuscript begins with an introduction written in clear and understandable language. The authors review the progress achieved in the treatment of oncological diseases over the past 20 years. Indeed, tyrosine kinase inhibitors have revolutionized. The median survival time has increased significantly, I know this not only from the literature, but also from the example of my family. Currently, tyrosine kinase inhibitors are used in combination with other effective therapeutic agents. It should be noted that tyrosine kinase inhibitors with anticancer nanoparticles are quite actively used, for example 10.3390/ijms23126641. Or, on the contrary, nanoparticles are made for targeted delivery of sorafenib to the tumor, for example 10.1016/j.jddst.2022.103142. All this needs to be taken into account in the manuscrip
Response 1: Thank you for pointing this out. We agree with this comment. We have supplemented the Introduction with these cites. [The changes can be found on Page 2, paragraph 6, line 79-87.]
Comments 2: The materials and methods section is written quite clearly, they took other people's data and analyzed it. The results section of the manuscript begins ambiguously. In section 3.1.1. there are references to literature in the title. How to understand this?
Response 2: These references are the same as those in the table presenting the datasets (Table 2). To make it more clear, we have removed the references from the titles.
Comments 3: Then the authors step by step consider each individual case from the database and build a visual graph based on the known data, and so on 14 times. After this, the manuscript contains a huge section of the discussion manuscript. The first two paragraphs are devoted to general information, then there are 3 tables: one with the names of the genes, the second and third with the levels of significance. Then there are 3 pages of text that have little to do with the information obtained in the results section…
Response 3: Thank you for pointing this out. We agree with your comment. We have done these modifications, so some parts of the discussion section were transferred to the results, and we’ve also deleted the unnecessary parts. [You can find on Page 5, Paragraph 1, Line 149-155; Page 6, Paragraph 1, Line 164-166, 169-175; Page 7, Paragraph 1, Line 202-210; Page 7, Paragraph 1, Line 219-225; Page 7, Paragraph 1, Line 235-240; Page 8, Paragraph 1, Line 251-255; Page 8, Paragraph 1, Line 263-267; Page 9, Paragraph 1, Line 300-304, Page 10, Paragraph 1, Line 306-313; Page 10, Paragraph 1, Line 322.334; Page 10, Paragraph 1, Line 345-355; Page 11, Paragraph 1, Line 364-371; Page 11, Paragraph 1, Line 378-379, 383-390; Page 11, Paragraph 1, Line 401-406; Page 12, Paragraph 1, Line 407-411; Page 13, Paragraph 1, Line 440-447, 453-455]
Comments 4: The main discovery of the manuscript is that tyrosine kinase inhibitors significantly affect the expression of epigenetic writers, erasers, miRNAs and members of chromatin remodeling complexes. Almost all tyrosine kinase inhibitors provide a decrease in tumor cell proliferation, which suggests that the authors have made the right conclusions, even if they were too expected…
Minor points: The manuscript states that this is a review, but it is a regular article. The manuscript is submitted to the journal Cancer, while the file I am reviewing says the journal Biomedicine.
Response 4: To make it clear, the manuscript has initially been submitted to Biomolecules based on an invitation regarding the Special Issue entitled Genetic and Genomic Biomarkers in Cancer Diagnosis, Prognosis, and Treatment Prediction. The editor of that special issue recommended the transfer of our manuscript to the esteemed journal Cancers, which is also of distinguished interest by our working group, where we have previously published our results about recent treatment outcome measures in AML, registered by the Hungarian Pediatric Oncology Group (HPOG) (PMID: 34680225). We absolutely agree with the Reviewer, that our manuscript is an original article, not a review.
Comments 5: Figures should be entered into the text after the first mention, and not after 3 pages. Constantly flipping back and forth through the manuscript is inconvenient!
Response 5: Thank you for pointing this out. We agree with your comment. We’ve done the asked modifications, so the Figures are now in the same Paragraphs where they first mentioned.
Comments 6: In vitro / in vivo and other Latin are written in italics
Response 6: Thank you for pointing this out. We’ve done the asked modifications, so now they are written in italics.
Reviewer 2 Report
Comments and Suggestions for Authors
Impact of tyrosine kinase inhibitors on the expression pattern of epigenetic regulators
Summary
In this paper, Tóth et al are interested in the impact of TKIs on epigenetic modulation in in vitro or in vivo tumor models. The approach is interesting, with a comparative analysis of exposed/unexposed and before/after. From a panel of 200 genes (GEO DATABASE), they observed the specific expression of genes that regulate epigenetics.
The idea is interesting, but this article lacks an analytical approach. The authors could limit themselves in the description of the number of models to focus on a few models with more signaling analysis.
Major comments
- The discussion is a repetition of the results. We need to give more explanations. It looks like a descriptive list with no explanatory analysis. Limited interest.
- In Table 3, which summarizes the differential expression, there is a need for an interpretive column to explain the role of each RNA identified as over- or under-expressed after exposure to TKI.
- The discussion should not be a repetition of the results, it should be much shorter and interpretive. Try to explain these differential expressions. Figures and tables should not be cited in the discussion/conclusion but in the results (e.g. for table 2).
- Could the authors discuss a potential interest of a transcriptomic look in this type of analysis?
Minor comments
- Page 2: Line 50-61: is it really necessary to detail all the indications? This weighs down the paper and does not add anything
- Page 13 line 459-469: not adapted in the discussion -> To replace in introduction
- Table 2 and comments -> to be put in the introduction
- Table 3 and comments : to put in results
Author Response
REVIEWER 2
Comments 1: In this paper, Tóth et al are interested in the impact of TKIs on epigenetic modulation in in vitro or in vivo tumor models. The approach is interesting, with a comparative analysis of exposed/unexposed and before/after. From a panel of 200 genes (GEO DATABASE), they observed the specific expression of genes that regulate epigenetics. The idea is interesting, but this article lacks an analytical approach. The authors could limit themselves in the description of the number of models to focus on a few models with more signaling analysis.
Response 1: We would like to sincerely thank the Reviewer for the insightful and thought-provoking comments. We fully agree with these observations, and to the best of our ability, we have incorporated the suggestions into the revised manuscript. We believe that these changes have made the manuscript more substantial and improved its overall quality.
Comments 2: The discussion is a repetition of the results. We need to give more explanations. It looks like a descriptive list with no explanatory analysis. Limited interest.
Response 2: Thank you for pointing this out. We agree with your comment. We have done these modifications, so some parts of the discussion section were transferred to the results, and we’ve also deleted the unnecessary parts. [You can find on Page 5, Paragraph 1, Line 149-155; Page 6, Paragraph 1, Line 164-166, 169-175; Page 7, Paragraph 1, Line 202-210; Page 7, Paragraph 1, Line 219-225; Page 7, Paragraph 1, Line 235-240; Page 8, Paragraph 1, Line 251-255; Page 8, Paragraph 1, Line 263-267; Page 9, Paragraph 1, Line 300-304, Page 10, Paragraph 1, Line 306-313; Page 10, Paragraph 1, Line 322.334; Page 10, Paragraph 1, Line 345-355; Page 11, Paragraph 1, Line 364-371; Page 11, Paragraph 1, Line 378-379, 383-390; Page 11, Paragraph 1, Line 401-406; Page 12, Paragraph 1, Line 407-411; Page 13, Paragraph 1, Line 440-447, 453-455; Page 5, Paragraph 1, Line 149-155] To make it more clear we’ve created a new Figure about the exact process of the data collection (Figure S3), you can find it in the supplementary section. We hope that this helps you to understand our findings more easily.
Comments 3: In Table 3, which summarizes the differential expression, there is a need for an interpretive column to explain the role of each RNA identified as over- or under-expressed after exposure to TKI.
Response 3: Thank you for your suggestion, we absolutely agree with it. We’ve added an extra column to Table 3, where we’ve marked in the case of each RNA identified as over- or under-expressed after exposure to TKI. We hope that it helps the understanding.
Comments 4: The discussion should not be a repetition of the results, it should be much shorter and interpretive. Try to explain these differential expressions. Figures and tables should not be cited in the discussion/conclusion but in the results (e.g. for table 2).
Response 4: Thank you for your suggestion. Results and Discussion have been restructured based on the suggestions of the Reviewer. In the revised version of the manuscript, all figures and tables are incorporated into the Introduction and Results. You can find on Page 3 (Table 1), Page 14 (Table 3).
Comments 5: Could the authors discuss a potential interest of a transcriptomic look in this type of analysis?
Response 5: Thank you very much for this excellent suggestion. We completely agree that incorporating a transcriptomic analysis could add an interesting layer to the study and provide further insights into the molecular mechanisms underlying our findings. While our current study focuses on a small subset of epigenetic regulators, we acknowledge that a transcriptomic approach would offer valuable complementary information, especially in understanding gene expression profiles and their correlations with the observed outcomes. We have taken note of this idea for future work, and it is certainly something we would consider exploring in subsequent studies to deepen the analysis.
Comments 6: Page 2: Line 50-61: is it really necessary to detail all the indications? This weighs down the paper and does not add anything.
Response 6: Thank you very much for this suggestion. We’ve deleted the mentioned Paragraph from the Introduction section. We hope that it affects the value of the Article positively.
Comments 7: Page 13 line 459-469: not adapted in the discussion -> To replace in introduction
Response 7: Thank you for your suggestion. We’ve restructured the Intorduction according to your suggestion. You can find on Page 2-3, Paragraph 7, Line 89-100.
Comments 8: Table 2 and comments -> to be put in the introduction
Response 8: Thank you for your suggestion. In the revised version of the manuscript, Table 2 was incorporated into the Introduction, which you can find as Table 1. You can find on Page 3 as Table 1.
Comments 9: Table 3 and comments : to put in results
Response 9: Thank you for your suggestion. Results and Discussion have been restructured based on the suggestions of the Reviewer. In the revised version of the manuscript, all figures and tables are incorporated into the Intorduction and Results. You can find on Page 14 (Table 3).
Reviewer 3 Report
Comments and Suggestions for Authors
In the manuscript "Impact of tyrosine kinase inhibitors on the expression pattern of epigenetic regulators" the authors provide an interesting presentation of the effects of tyrosine kinase inhibitors from the angle of the expression of epigenetic regulators. TKI influence the expression of epigenetic regulators substantially, according to this study. The methodology is clearly outlined and the results are presented with the crucial addition of figures.
Figure 2 a, there seems to be an incomplete margin, which cuts the x-axis off the graphs for miR382, miR516B1 and miR664B
The subtitle "6. Patents" seems to be there with no text, as it is followed by the listing of Supplements. Perhaps can be deleted.
The listing of Supplements mentions that they can be downloaded, even though they are included in the Appendix to the full text.
Author Response
REVIEWER 3
Comments 1: In the manuscript "Impact of tyrosine kinase inhibitors on the expression pattern of epigenetic regulators" the authors provide an interesting presentation of the effects of tyrosine kinase inhibitors from the angle of the expression of epigenetic regulators. TKI influence the expression of epigenetic regulators substantially, according to this study. The methodology is clearly outlined and the results are presented with the crucial addition of figures.
Response 1: We would like to sincerely thank the Reviewer for the insightful and thought-provoking comments. We fully agree with these observations, and to the best of our ability, we have incorporated the suggestions into the revised manuscript. We believe that these changes have made the manuscript more substantial and improved its overall quality.
Comments 2: Figure 2 a, there seems to be an incomplete margin, which cuts the x-axis off the graphs for miR382, miR516B1 and miR664B.
Response 2: Thank you for your remark, we couldn’t see the incomplet margin on these Figures.
Comments 3: The subtitle "6. Patents" seems to be there with no text, as it is followed by the listing of Supplements. Perhaps can be deleted.
Response 3: Thank you for your suggestion. We’ve corrected it, it was just a little mistake, that escaped our attention. Now you can read the article without the 6. Patents.
Comments 4: The listing of Supplements mentions that they can be downloaded, even though they are included in the Appendix to the full text.
Response 4: Thank you for pointing this out. We’ve deleted the Appendix section with the Supplementary figures. They can be just downloaded. We hope that it improves the structure of the article.
Reviewer 4 Report
Comments and Suggestions for Authors
Klaudia Toth, and Zsuzsanna Gaal have wrote an excellent review on "Impact of TKI........ regulators". TKI inducing epigenetic regulators is an observation which can help in longer term as new epigenetic drugs and therapies are emerging. The review is well written, and I have following comments-
1 It would be helpful for readers if a table is added summarizing all the types of epigenetic regulators observed in the review and if any drugs (clinical or under trial) available against the epigenetic regulator.
2 Authors can add a cartoon diagram summarizing the theme/mechanism of the observation in the paper.
3 Authors can include the molecular mechanism known/proposed, why and how TKI affects epigenetic regulators (mainly observed in this review)
Author Response
REVIEWER 4
Comments 1: Klaudia Toth, and Zsuzsanna Gaal have wrote an excellent review on "Impact of TKI........ regulators". TKI inducing epigenetic regulators is an observation which can help in longer term as new epigenetic drugs and therapies are emerging. The review is well written, and I have following comments.
Response 1: We would like to sincerely thank the Reviewer for the insightful and thought-provoking comments. We fully agree with these observations, and to the best of our ability, we have incorporated the suggestions into the revised manuscript. We believe that these changes have made the manuscript more substantial and improved its overall quality.
Comments 2: It would be helpful for readers if a table is added summarizing all the types of epigenetic regulators observed in the review and if any drugs (clinical or under trial) available against the epigenetic regulator.
Response 2: Thank you for your suggestion. Introduction, Results and Discussion have been restructured based on the suggestions of the Reviewer. In the revised version of the manuscript, all figures and tables are incorporated into the Intorduction and Resulta. Table 2 was incorporated into the Introduction, which you can find as Table 1 [Page 3].
We’ve added two extra columns to Table 3, where we’ve marked in the case of each RNA identified as over- or under-expressed after exposure to TKI and another one, in which we have marked if any drugs (clinical or under trial) are available against the epigenetic regulator with the clinical trial numbers. We hope that it helps the understanding. You can find on Page 14 (Table 3).
Comments 3: Authors can add a cartoon diagram summarizing the theme/mechanism of the observation in the paper.
Response 3: Thank you for pointing this out. We agree with your comment. To make it more clear we’ve created a new Figure about the exact process of the data collection (Figure S3), you can find it in the supplementary section. We hope that this helps you to understand our findings more easily.
Comments 4: Authors can include the molecular mechanism known/proposed, why and how TKI affects epigenetic regulators (mainly observed in this review)
Response 4: Thank you for this excellent suggestion. We completely agree that a deeper exploration of the molecular mechanisms underlying the effects of TKIs on epigenetic regulators would be highly valuable. However, as you rightly pointed out, there is currently limited data available on this specific interaction. We have highlighted the expression levels of epigenetic regulators in response to TKI treatments in our manuscript, but the precise molecular pathways remain an area in need of further investigation.
We are planning to explore this molecular relationship in more detail in future work, as we believe this will not only enhance our understanding of the disease pathogenesis but also open up potential new translational and clinical applications.
Round 2
Reviewer 1 Report
Comments and Suggestions for Authors
I hope I helped the authors improve the text of the manuscript.
Reviewer 2 Report
Comments and Suggestions for Authors
Corrections accepted for publication